# CD115− monocytic myeloid-derived suppressor cells are precursors of OLFM4high polymorphonuclear myeloid-derived suppressor cells

Yunyun Zou [1,2], Nobuhiko Kamada[3], Seung-Yong Seong [1,2✉] & Sang-Uk Seo [4✉]

Myeloid-derived suppressor cells (MDSCs) consist of monocytic (M-) MDSCs and polymorphonuclear (PMN-) MDSCs that contribute to an immunosuppressive environment in tumor-bearing hosts. However, research on the phenotypic and functional heterogeneity of MDSCs in tumor-bearing hosts and across different disease stage is limited. Here we subdivide M-MDSCs based on CD115 expression and report that CD115− M-MDSCs are functionally distinct from CD115+ M-MDSCs. CD115− M-MDSCs increased in bone marrow and blood as tumors progressed. Transcriptome analysis revealed that CD115− M-MDSCs expressed higher levels of neutrophil-related genes. Moreover, isolated CD115− M-MDSCs had higher potential to be differentiated into PMN-MDSCs compared with CD115+ M-MDSCs. Of note, CD115− M-MDSCs were able to differentiate into both olfactomedin 4 (OLFM4)hi and OLFM4lo PMN-MDSCs, whereas CD115+ M-MDSCs differentiated into a smaller proportion of OLFM4lo PMN-MDSCs. In vivo, M-MDSC to PMN-MDSC differentiation occurred most frequently in bone marrow while M-MDSCs preferentially differentiated into tumor-associated macrophages in the tumor mass. Our study reveals the presence of previously unrecognized subtypes of CD115− M-MDSCs in tumor-bearing hosts and demonstrates their cellular plasticity during tumorigenesis.

[1] Department of Biomedical Sciences, Seoul National University College of Medicine, Seoul, Republic of Korea. [2] Wide River Institute of Immunology, Seoul National University College of Medicine, Hongcheon, Republic of Korea. [3] Division of Gastroenterology and Hepatology, Department of Internal Medicine, University of Michigan, Ann Arbor, MI, USA. [4] Department of Microbiology, College of Medicine, The Catholic University of Korea, Seoul, Republic of Korea. ✉email: seongsy@snu.ac.kr; suseo@catholic.ac.kr

The complex etiology underlying tumorigenesis makes it difficult to develop an efficient treatment for cancer despite long-term and multifaceted efforts. The immunosuppressive network emerging from primary tumor results in the disruption of effector T cell responses and functional alteration of myeloid populations[1]. Neutrophils and monocytes are early responding myeloid cell populations and they play a pivotal role in inflammatory, anti-microbial, and wound-healing processes upon innate immune stimulus. But in the cancerous condition, neutrophils and monocytes are systemically shaped into immunosuppressive PMN-MDSCs and M-MDSCs, respectively, which enhance tumor growth[2]. In mice, the surface markers used for PMN-MDSC ($CD11b^+Ly6G^+Ly6C^{lo}$) and M-MDSC ($CD11b^+Ly6G^-Ly6C^{hi}$) overlap with steady-state counterparts but functionally play different roles[3,4]. Unlike M-MDSCs, PMN-MDSCs have surface marker lectin-type oxidized LDL receptor-1 (LOX-1) that distinguish them from non-suppressive PMN cells in humans[5,6]. OLFM4, an olfactomedin-related glycoprotein located in nuclear and mitochondria[7,8], it marks a subset of neutrophils in mice (7–35%) and human (20–25%)[9,10].

All MDSCs are derived from common myeloid precursors and require colony-stimulating factors (CSFs), such as GM-, G-, and M-CSF to achieve successful expansion, survival, and immunosuppressive activation[11,12]. CSFs also drive both monocytes and M-MDSCs (hereafter referred to as monocytic cells) to migrate to the target tissue[11]. Monocytic cells have a short life span, but some participate in the prolonged immune response by further differentiation after migration[12]. Unlike monocytes, which are precursors of tissue-resident macrophages and dendritic cells (DCs)[13], M-MDSCs migrate to the tumor site and differentiate into tumor-associated macrophages (TAMs) and inflammatory DCs (inf-DCs)[14]. These tumor-associated cells augment immunosuppression in the microenvironment and crosstalk with other infiltrated immune cells to promote tumor growth and metastasis[14]. This process is regulated by CSFs and has multiple signaling pathways (e.g., Ras/MAPK, PI3K/Akt, Jak/Stat, and TGFβ)[15]. As a central regulator in multiple steps of MDSC development, CSFs recognized by their cognate receptors are potential targets for cancer therapy[16,17].

At least two phenotypically distinct monocyte-derived macrophage subtypes are found within the tumor microenvironment (TME)[18–20]. M1 TAM plays an anti-tumor role and requires GM-CSF (CSF2) and pro-inflammatory stimuli (e.g., TNF-α) for generation. Whereas, M2 TAM is immunosuppressive, contributes to tumor growth, and requires M-CSF (CSF1) and anti-inflammatory stimuli (e.g., IL-4) for generation[19,21]. M-CSF exclusively signals through its unique receptor, CD115 (CSF1R), while CD115 has another ligand, IL-34[22]. Generally, IL-34 is mainly expressed by neurons and keratinocytes in the brain and skin, whereas M-CSF is produced by multiple cell types in a broad range of tissues (e.g., fibroblasts, macrophages, and tumor cells)[23,24]. Therefore, M-CSF may be the major player in CD115 signaling in TAM differentiation of tumor-bearing (TB) animals. Since M2 TAM differentiation relies on M-CSF and CD115 is a traditional phenotypic marker for monocytes, CD115 becomes a promising target for cancer therapy[25,26]. The depletion of M2 TAM by using CD115 inhibitors (e.g., BLZ945, PLX3397, and RG7155) has been investigated in both animal tumor models and in clinical trials, but the results were mixed, ranging from moderate to less than satisfactory[20,23,27,28]. In fact, tumor-driven M-CSF activates CD115 signaling to down-regulate the secretion of granulocytic chemokine secretion by cancer-associated fibroblasts (CAF) and restrict PMN-MDSC infiltration[29]. Thus, CD115 blockade eliminates the M-CSF driven M2 TAM while attracting more pro-tumoral PMN-MDSCs, which mitigate the therapeutic effect of the CD115 inhibitor[29]. This suggests that CD115 signaling plays an important role in balancing TAM and PMN-MDSCs in the TME.

Accumulating evidence indicates that intra-tumoral cell populations have high plasticity and close contact or crosstalk[2,3,18,29]. These complex interactions make it challenging to develop efficient cancer therapies that targeting only one branch of immunosuppressive cells (e.g., PMN-MDSC, M-MDSC, and TAM). Therefore, further efforts for in-depth investigation of the plasticity and heterogeneity of immunosuppressive populations are needed. Among immunosuppressive populations, M-MDSCs have the highest plasticity and differentiation potency[3]. In the current study, we attempted to dissect a subset of the M-MDSC population in TB mice based on their surface expression of CD115 and address the multi-lineage differentiation potential of M-MDSC subsets. $CD115^-$ M-MDSCs were part of the monocytic cell population in three subcutaneous tumor models (EL4, LLC1, and MC38) and expanded as tumor progressed in the host. The functional studies and transcriptome analysis demonstrated that $CD115^-$ and $CD115^+$ M-MDSCs are distinct monocytic populations. We also demonstrated that $CD115^-$ M-MDSCs differentiate into both $OLFM4^{hi}$ PMN-MDSCs and $OLFM4^{lo}$ PMN-MDSCs, whereas $CD115^+$ M-MDSCs exclusively differentiate into a smaller portion of $OLFM4^{lo}$ PMN-MDSCs.

## Results

**$CD115^-$ M-MDSCs accumulate in TB mice.** Both monocytes in naive mice and M-MDSCs in TB mice commonly express CD11b and Ly6C on their surfaces. We used these markers for flow cytometry analysis along with other cell type-specific markers to exclude other leukocytes: CD11c for DCs, SiglecF for eosinophils, and Ly6G for neutrophils (Supplementary Fig. 1a). To assess the expression of the classical phenotypic monocyte marker CD115 on M-MDSCs, we subcutaneously injected mice with tumor cells (EL4, LLC1, and MC38). Tumor implantation triggered alteration in bone marrow (BM) and blood cellularity (Fig. 1a). As expected, EL4 injection significantly increased blood $CD115^+$ M-MDSCs (Supplementary Fig. 1b). But contrary to our expectation that all monocytic cells ($CD11c^-CD11b^+Ly6G^-SiglecF^-Ly6C^{hi}$) would express CD115, we found that a subset of monocytic cells did not express CD115 in either naive or TB mice (Fig. 1b). $CD115^-$ monocytic cells isolated from naive or EL4 TB mice had the characteristic morphology of monocytes including large reniform nuclei (Fig. 1c). Remarkably, the proportion and the absolute number of $CD115^-$ M-MDSCs were increased in the BM of EL4 TB mice compared to naive mice while $CD115^+$ M-MDSCs were relatively decreased (Fig. 1d). $CD115^-$ M-MDSCs were also increased in LLC1 and MC38 TB mice, suggesting a positive correlation between the increased frequency of $CD115^-$ M-MDSCs and tumorigenesis (Fig. 1d). The proportions of $CD115^-$ M-MDSCs were comparable between male and female mice (Supplementary Fig. 1c). As $CD115^-$ M-MDSCs accumulated in the BM, $CD115^-$ M-MDSCs also increased in the blood circulation (Fig. 1e). We also performed a time-course analysis of $CD115^-$ M-MDSC populations in MC38 TB mice. Total cell numbers in BM and blood were gradually increased after MC38 implantation but less prominent compared to EL4 (Supplementary Fig. 2a). Meanwhile, the ratio and total $CD115^-$ M-MDSCs in the MC38 TB model increased less compared to EL4 (Supplementary Fig. 2b, c). Our data demonstrate that M-MDSCs have the $CD115^-$ subset in TB mice and their expansion profile varies by type of cancer and stage of tumorigenesis.

**$CD115^-$ and $CD115^+$ M-MDSCs are functionally distinct.** Next, we isolated $CD115^-$ and $CD115^+$ monocytic cells from BM

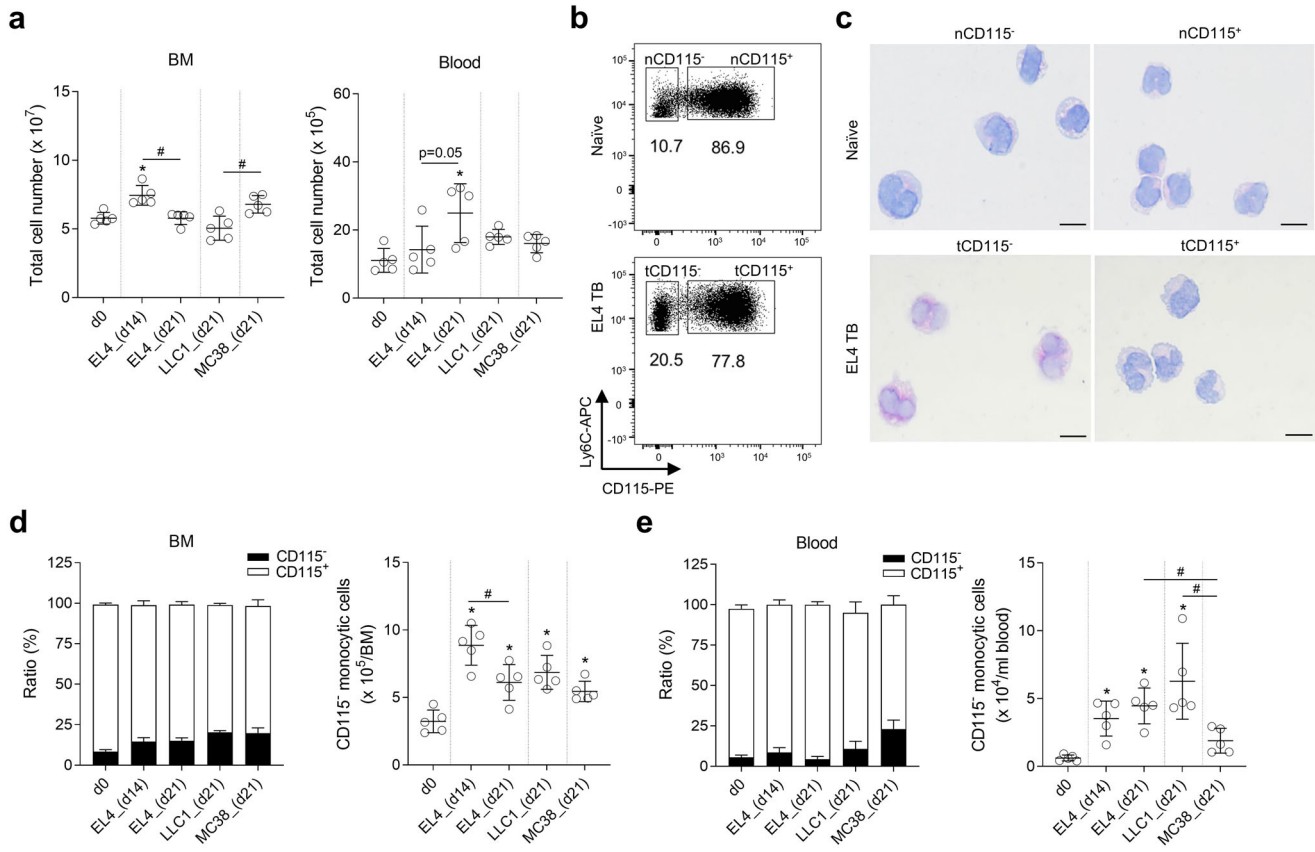

**Fig. 1 TB mice have increased levels of CD115$^-$ M-MDSCs in BM. a** Total BM and blood cell counts in naive and EL4, LLC1, and MC38 TB mice at indicated days after tumor implantation ($n = 5$ per group). **b** Representative plot shows CD115$^-$ and CD115$^+$ monocytic cells (CD11c$^-$CD11b$^+$Ly6G$^-$SiglecF$^-$Ly6C$^{hi}$) in naive and EL4 TB mice. The letter "n" and "t" in front of CD115 indicate that cells were isolated from tumor-free and TB mice, respectively. **c** Representative images of H&E stained CD115$^-$ and CD115$^+$ cells from BM of naive or EL4 TB mice. Scale bars = 10 μm. **d, e** Relative ratios of CD115$^-$ and CD115$^+$ M-MDSCs and absolute numbers of CD115$^-$ monocytic cells in BM (**d**) and blood (**e**) of EL4, LLC1, and MC38 TB mice ($n = 4$ or 5 in each group). One-way ANOVA with correction for multiple comparisons test was used: *$p < 0.05$ (all vs. d0); #$p < 0.05$. Data are mean ± SD.

of naive or EL4 TB mice to evaluate their immunosuppression on lymphocyte proliferation. CD8$^+$ T cells isolated from spleens of naive mice were stimulated with anti-CD3/CD28 beads to proliferate and sorted monocytic cells were co-cultured at different ratios. Although CD115$^-$ M-MDSCs suppressed the proliferation of CD8$^+$ T cells more efficiently than CD115$^+$ M-MDSCs, both CD115$^-$ and CD115$^+$ M-MDSCs showed comparable inhibition of IFN-γ secretion (Fig. 2a–c). As expected, both CD115$^-$ and CD115$^+$ monocytes from naive mice did not show suppressive phenotypes (Supplementary Fig. 3a, b). We also assessed the phagocytosis capacity of the monocytic cells by co-culturing them with FITC-labeled *Escherichia coli*. CD115$^-$ M-MDSCs from BM and blood showed a stronger phagocytosis capacity than CD115$^+$ M-MDSCs (Fig. 2d). Immune cells under tumor conditions face the hypoxic microenvironment that induces oxidative stress[30]. In order to test resistance to oxidative stress, we exposed CD115$^-$ and CD115$^+$ M-MDSCs to various amounts of hydrogen peroxide ($H_2O_2$) for 2 h. CD115$^-$ M-MDSCs exhibited higher resistance to increasing oxidative stress conditions than CD115$^+$ M-MDSCs (Fig. 2e). CD115$^-$ M-MDSCs also showed significantly lower spontaneous apoptosis without exposure to $H_2O_2$. Similarly, CD115$^-$ monocytes from naive mice were stronger in phagocytosis and more resistant to oxidative stress than CD115$^+$ monocytes (Supplementary Fig. 3c, d).

Pre-metastatic niche formation is recognized as a sign of a metastasis-promoting environment predating metastasis[31]. As bone marrow-derived cells, including CD11b$^+$ myeloid cells,

promote metastasis of tumor cells[32], we examined the migration and invasion of EL4 or LLC1 cells in the presence of conditioned medium (CM) that was prepared from CD115$^-$ or CD115$^+$ M-MDSC culture (Supplementary Fig. 3e). Both CM from CD115$^-$ or CD115$^+$ M-MDSCs significantly enhanced migration and invasion of the EL4 and LLC1 cells compared to control medium without CMs while the effect of CM from CD115$^-$ and CD115$^+$ M-MDSCs was comparable.

At the primary tumor site, TAMs contribute to creating a suitable microenvironment by secreting many cytokines (e.g., TGF-β) or crosstalk with other cell types (e.g., CAF) that accelerate the tumor cell invasion and metastasis[33]. To compare TGF-β secretion by CD115$^-$ and CD115$^+$ M-MDSCs, we isolated M-MDSCs from TB mice and measured TGF-β in CM. CD115$^+$ M-MDSCs isolated from EL4 TB mice showed increased tendency of TGF-β secretion compared to CD115$^-$ M-MDSCs but the difference did not reach statistical significance. No difference was observed in CM from LLC1 TB mice (Supplementary Fig. 3f). Based on our data, CD115$^-$ subpopulations in mouse tumor models are immunosuppressive MDSCs and capable of promoting metastasis of cancer cells. However, CD115$^-$ M-MDSCs had a greater degree of T cell suppression than CD115$^+$ M-MDSCs.

**CD115$^-$ and CD115$^+$ M-MDSCs have distinct gene expression profiles.** In addition to a functional assessment, we performed

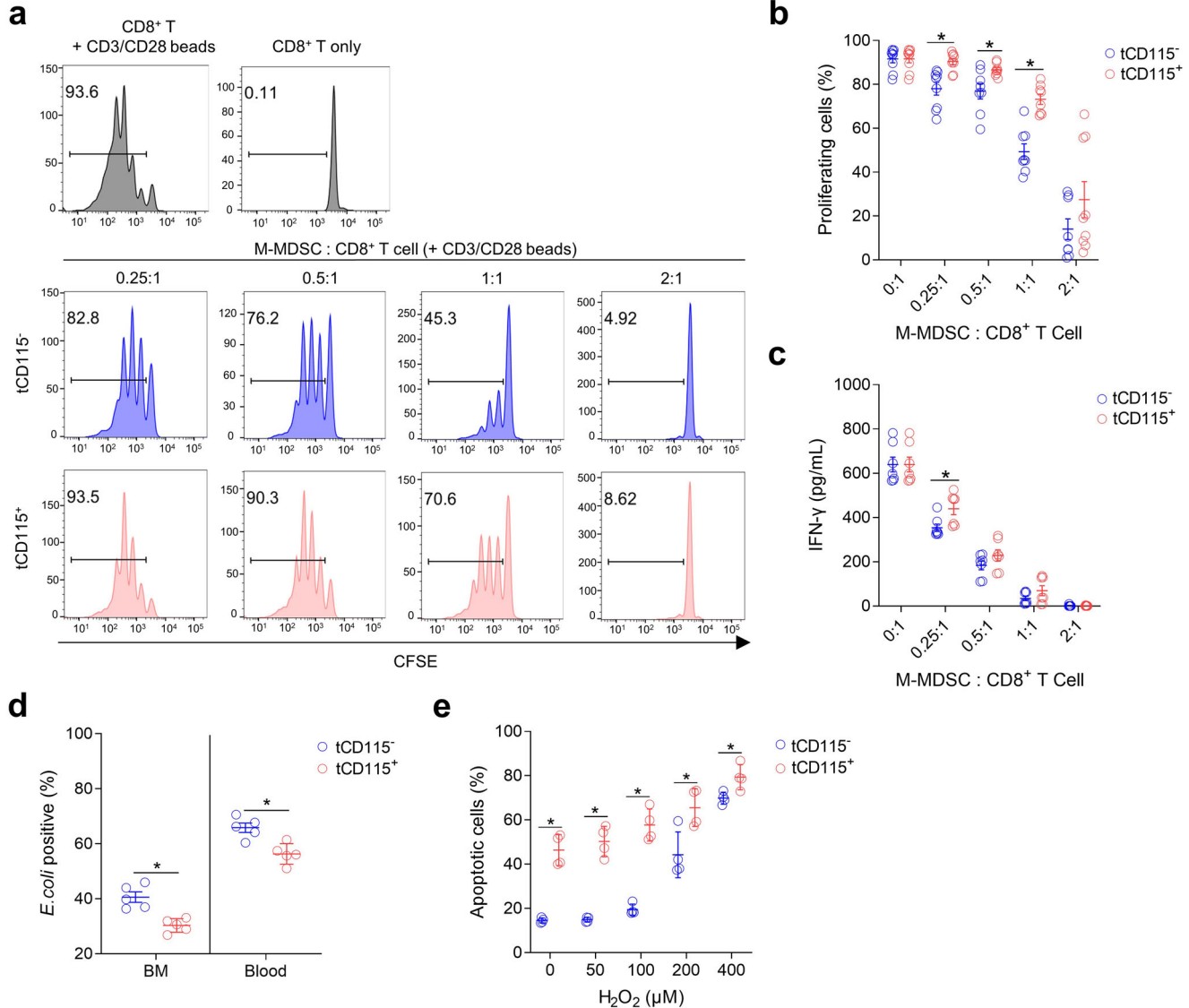

**Fig. 2 CD115⁻ M-MDSCs have distinct phenotypes from CD115⁺ M-MDSCs. a–c** Resting CD8⁺ T cells were activated with anti-CD3/CD28 beads in the presence of indicated ratios of sorted CD115⁻ or CD115⁺ M-MDSCs isolated from BM of EL4 TB mice. Suppression of CD8⁺ T cell proliferation was assessed 3 days after co-culture. **a** Representative FACS histogram. **b** Ratios of proliferating CD8⁺ T cells ($n = 8$–9, data are pooled from three independent experiments). **c** IFN-γ levels in culture supernatant ($n = 8$, data are pooled from three independent experiments). **d** BM and blood cells from EL4 TB mice ($n = 5$ each) were incubated with FITC-labeled *E. coli* and the ratios of FITC-positive CD115⁻ and CD115⁺ M-MDSCs were assessed by FACS. **e** Total BM cells were harvested from EL4 TB mice ($n = 4$) and treated with indicated concentrations of $H_2O_2$ for 2 h to induce apoptosis. Annexin-V single positive and Annexin-V/7-AAD double-positive cells were assessed from CD115⁻ and CD115⁺ M-MDSCs. Data are pooled from two independent experiments. Multiple unpaired Student's *t* test: *$p < 0.05$. Data are mean ± SD.

RNA sequencing (RNA-Seq) analysis to compare global gene expression between CD115⁻ and CD115⁺ M-MDSCs. Principal component analysis (PCA) showed a clear difference in gene expression between CD115⁻ and CD115⁺ subsets with 77% of the variation explained by the principal component (PC) 1 (Fig. 3a). In addition, 15% of the variation was related to PC2, which showed differential gene expression between EL4 TB MDSCs and naive monocyte counterparts (Fig. 3a). The transcriptome difference of monocytic cells was confirmed by a hierarchical clustering tree (Fig. 3b). When the cutoff value was set at 2-fold, differentially expressed genes (DEGs) between CD115⁻ and CD115⁺ subsets of M-MDSCs overlapped in only 1.8% (20/1107) of EL4 TB mice (Fig. 3c). CD115⁻ and CD115⁺ subsets of naive monocytes also showed marginal overlap between two subsets (Supplementary Fig. 4a). Likewise, when

CD115⁻ or CD115⁺ subsets were compared between naive or EL4 TB mice, they also showed little overlap, indicating that all gene transcriptions are distinct in each monocytic subset (Supplementary Fig. 4a).

In all, 1,723 DEGs were grouped in six clusters (Fig. 3d). DEGs of interest were selected in gene ontology terms related to granulocyte, transcription factors, mononuclear phagocytes, and angiogenesis (Fig. 3e, Supplementary Fig. 4b–d). Among listed transcript factor DEGs, *Cebpe* (essential transcription factors required for granulocytic differentiation)[34] and *Erg* (essential for macrophage differentiation)[35] were highly expressed in CD115⁻ M-MDSCs compared to CD115⁺ M-MDSCs (Supplementary Fig. 4b). Also, several monocyte/macrophage-related DEGs were significantly increased in CD115⁻ M-MDSC (Supplementary Fig. 4c). These included *Chil3/Ym1* (the M2 macrophage

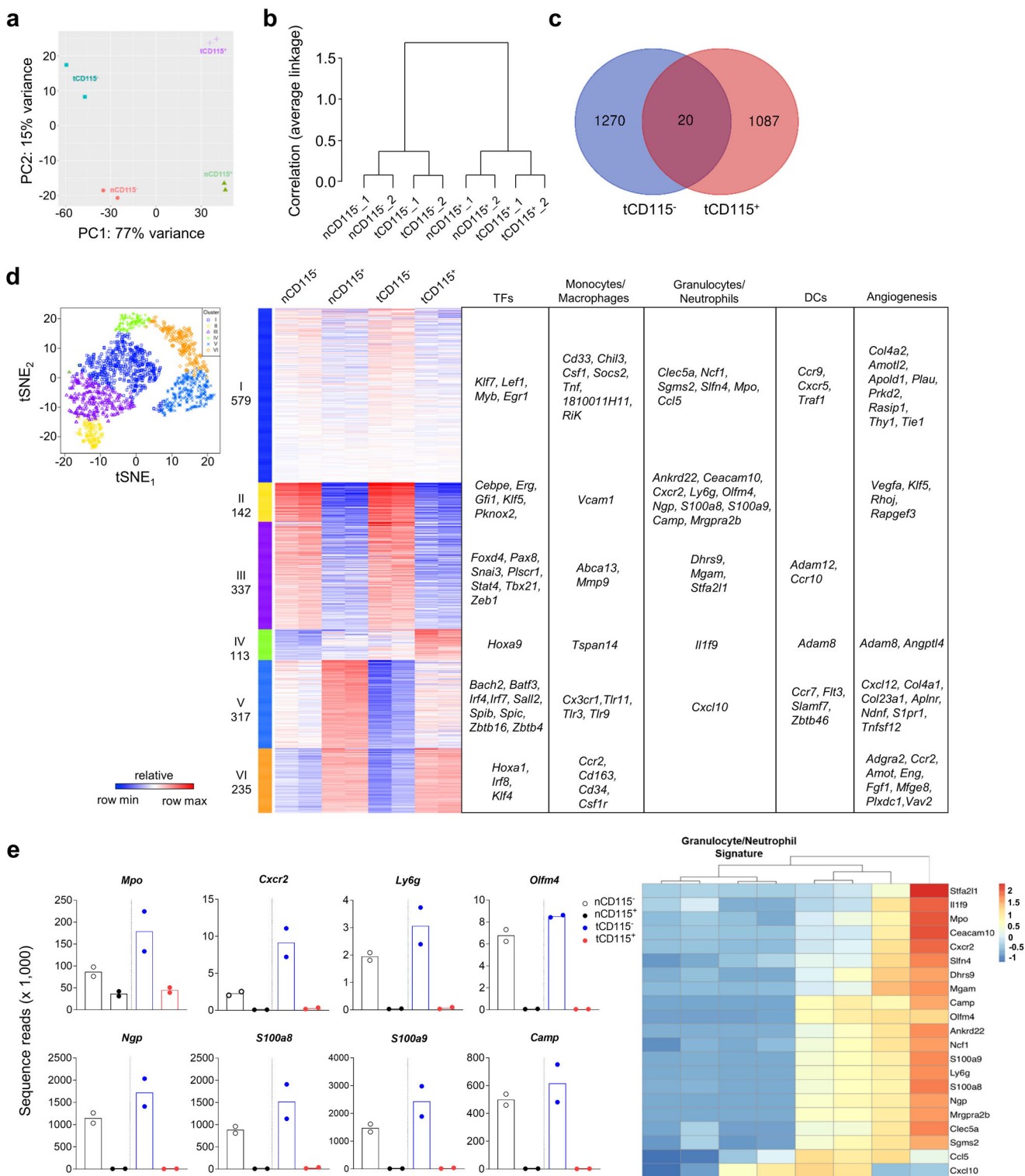

**Fig. 3 Granulocyte/neutrophil-related gene expression in CD115⁻ monocytic cells.** CD115⁻ and CD115⁺ monocytic cells were sorted from BM of naive or EL4 TB mice. Total RNA was extracted and subjected to transcriptome analysis. Data shown are from two biological replicates of each group. The letter "n" and "t" in front of CD115 indicate that cells were isolated from tumor-free and TB mice, respectively. **a** Principal component analysis plot. **b** Hierarchical cluster of four cell types. **c** Venn diagram shows variation between sorted CD115⁻ and CD115⁺ M-MDSC libraries in EL4 TB mice. **d** t-SNE plot of K-means clusters and lists of representative genes with more than 2-fold changes. DCs, dendritic cells. TFs, transcription factors. **e** Sequence reads and heatmap of selected granulocyte/neutrophil-associated variable genes (each dot in graph represents different biological replicates).

phenotypic marker)[36], *Csf1* (which accelerates differentiation and maturation of monocytes)[25], and *Tnf* (which is associated with macrophage and DC differentiation)[37]. CD115[+] M-MDSCs showed greater expression of *Cx3cr1*, *Ccr2*, and *Cd163* (M2 macrophage phenotype markers)[25] as did *Csf1r* (Supplementary Fig. 4c). Also, the analysis showed that seven angiogenesis-related genes are highly expressed in CD115[−] M-MDSCs, including *Vegfa* (Supplementary Fig. 4d). Of most interest, granulocyte/ neutrophil-related DEGs (e.g., *Mpo*, *Cxcr2*, *Ly6g*, *Olfm4*, *Ngp*, *S100a8*, *S100a9*, *and Camp*) were expressed at higher rates in CD115[−] subsets from both naive and EL4 TB mice (Fig. 3e). These data suggest that the CD115[−] subset may have a closer relationship with neutrophils or PMN-MDSCs than the CD115[+] subsets. Also, our overall transcriptomic analysis further suggests that CD115[−] M-MDSCs and monocytes are separate subsets of monocytic cells.

**CD115[−] M-MDSCs differentiate into PMN-MDSCs more efficiently outside the tumor mass.** In disease-free conditions, monocytes differentiate into macrophages and DCs[13]. M-MDSCs can also differentiate into PMN-MDSCs in the tumor host[38]. Because of the substantial expression of granulocytic-related genes in CD115[−] M-MDSCs, we assessed whether this M-MDSC subpopulation is prone to PMN-MDSC differentiation. BM CD115[−] and CD115[+] M-MDSCs were sorted from EL4 TB mice and labeled with CFSE. Each subset was transferred to other EL4 TB recipient mice and CFSE[+] cells were analyzed 48 h after the transfer (Fig. 4a). Compared to CD115[+] M-MDSCs, CD115[−] M-MDSCs had more frequent PMN-MDSC (CD11b[+]Ly6G[+]Ly6C[lo]) differentiation in BM, blood, and liver, but not in the tumor mass (Fig. 4b). Notably, more than 40% of CD115[−] M-MDSCs were differentiated into PMN-MDSCs in the BM (Fig. 4b). In the tumor mass, the lowest proportion of PMN-MDSCs (<6%) was found among samples from CD115[−] M-MDSC-transferred mice (Fig. 4b), whereas more than 70% of both CD115[−] and CD115[+] M-MDSCs were differentiated into F4/80[+] TAM (Fig. 4c). More than half the TAMs were CD206[+], indicating the majority of macrophages in the tumor mass are pro-tumoral M2 phenotype (Fig. 4c). M-MDSCs also differentiated into macrophages in other organs but were mostly CD206[−] (Supplementary Fig. 5a). We also tested the LLC1 tumor model and obtained results consistent with those in EL4 mice (Supplementary Figs. 5b and 6a–d).

In vitro, we cultured CD115[−] and CD115[+] M-MDSCs in the presence of GM-CSF with or without EL4 TES. Although GM-CSF-driven PMN-MDSC differentiation was less frequent in vitro, CD115[−] M-MDSCs were more efficient than CD115[+] M-MDSCs (Fig. 4d). When TES was supplemented in the culture medium to mimic the tumor microenvironment, PMN-MDSC differentiation from CD115[−] M-MDSCs was slightly reduced while M2 TAM differentiation was significantly increased (Fig. 4d), consistent with the in vivo data (Fig. 4b, c). Remarkably, CD115[−] M-MDSCs generated a lower proportion of M2 TAM (~65%) compared to CD115[+] M-MDSCs (~92%). Also, the morphology of macrophages further proved that TES drives M1 (round-shaped) to M2 (spindle-shaped) polarization. Although CD115[−] M-MDSCs differentiated to PMN-MDSCs less frequently in vitro, mature PMN-MDSCs were observed in CD115[−] M-MDSC culture with TES (Fig. 4e). Similar in vitro differentiation patterns were seen using LLC1 TES (Supplementary Fig. 6e, f). Collectively, our findings indicate that CD115[−] M-MDSCs can play a role as PMN-MDSC precursors in the outside of tumor environment but preferentially differentiate into M2 TAM in the tumor microenvironment.

**CD115[−] M-MDSCs differentiate into OLFM4[hi] PMN-MDSCs.** OLFM4 expression is used to define a subset of neutrophils in mice and humans[9,10]. However, the differentiation of OLFM4[hi] neutrophils is not completely understood. Because the *Olfm4* gene was highly expressed in the CD115[−] M-MDSCs (Fig. 3e), we further evaluated the correlation between CD115[−] M-MDSCs and OLFM4[hi] PMN-MDSCs in the TB hosts. To confirm the RNA-Seq data, we assessed OLFM4 expression in M-MDSCs and PMN-MDSCs by immunohistochemistry (IHC) (Fig. 5a). In EL4 TB mice, approximately 8% of PMN-MDSCs were OLFM4[hi] (Fig. 5a). Among the M-MDSCs, 34% of CD115[−] M-MDSCs expressed OLFM4 but fewer than 1% of CD115[+] M-MDSCs expressed OLFM4 (Fig. 5a). We labeled CD115[−] and CD115[+] M-MDSCs with CFSE and transferred to EL4 TB recipient mice to analyze OLFM4 expression in M-MDSC-derived cells (Fig. 5b). CD115[−] M-MDSC recipient mice had both CFSE[+] (donor origin) and CFSE[−] (recipient origin) PMN-MDSCs. As expected, 37% of CFSE[+] PMN-MDSCs expressed OLFM4, a percentage similar to that of CD115[−] M-MDSCs (Fig. 5c). When CFSE[+] cells were analyzed in CD115[+] M-MDSC recipient mice, OLFM4[hi] PMN-MDSCs were undetectable (Fig. 5c). The CFSE[+] cell tracking assay demonstrated that CD115[−] M-MDSCs have an inherent potential to generate both OLFM4[hi] and OLFM4[lo] PMN-MDSCs, whereas CD115[+] M-MDSCs only generate OLFM4[hi] PMN-MDSCs.

**Depletion of CD115[+] M-MDSC does not affect OLFM4[hi] PMN-MDSC.** Next, we adopted an in vivo depletion system for CD115[+] M-MDSCs (MM[DTR] mice) to confirm that CD115[+] M-MDSCs are not involved in OLFM4[hi] PMN-MDSC generation. Intraperitoneal injection of DT transiently depleted CD115[+] monocytic cells in MM[DTR] mice (Supplementary Fig. 7a). When these mice were given DT three times at 2-day intervals starting 7 days after EL4 tumor implantation (Supplementary Fig. 7b), the overall proportion of M-MDSCs in BM was reduced and the relative abundance of CD115[−] M-MDSCs was increased over CD115[+] M-MDSCs 14 days after tumor implantation (Fig. 6a). DT treatment significantly reduced tumor burden in MM[DTR] TB mice compared to C57BL/6 TB mice (Fig. 6b), indicating that CD115[+] M-MDSCs and the descendant cells contribute to tumor progression. Percentages and absolute numbers of PMN-MDSCs in BM were reduced in TB MM[DTR] mice upon DT treatment (Fig. 6c). This may be due to attenuated tumor burden or loss of CD115[+] M-MDSCs, the precursors of OLFM4[lo] PMN-MDSCs. Although TB mice significantly downregulated expression of neutrophil-related genes (*MPO*, *Olfm4*, *MMP9*, *NGP*, and *CAMP*) compared to mice without tumors (Fig. 6d), *Olfm4* expression levels were not affected by DT treatment in any mouse group (Fig. 6e). We used IHC to confirm that the proportion of OLFM4[hi] PMN-MDSCs was not affected by DT treatment in BM of TB MM[DTR] mice (Fig. 6f). We also repeated the experiment using C57BL/6 mice to exclude the effect of DT treatment. Unlike MM[DTR] mice, no changes in M-MDSCs and PMN-MDSCs were observed in C57BL/6 mice following DT treatment (Supplementary Fig. 8). Overall, our data suggest that depletion of CD115[+] M-MDSCs does not affect OLFM4[hi] PMN-MDSCs in TB mice.

**Discussion**
Cell surface molecules are useful to dissect heterologous myeloid cell populations. We used CD115 to identify a distinct subset of monocytic cells. When granulocyte-monocyte progenitors differentiate into common monocyte progenitors (cMoPs), they up-regulate CD115 expression on their surface and further develop into mononuclear phagocytes depending on M-CSF[39,40]. As

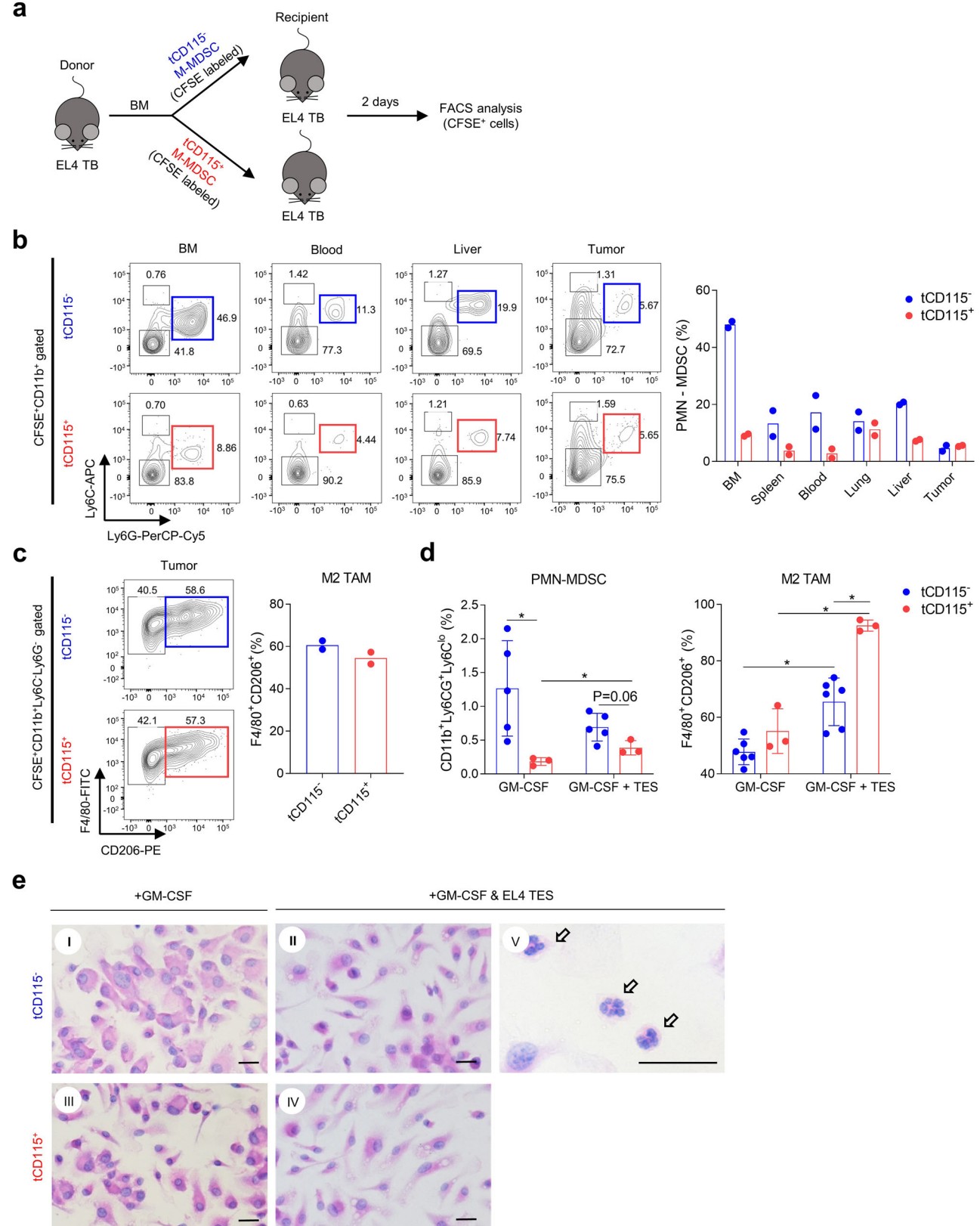

CD115 has long been used as a solid surface marker for monocytes and macrophages in both mice and humans[41], we tried to confirm the monocytic characteristics of CD115⁻ M-MDSCs. Despite loss of CD115 expression, these cells microscopically resemble typical monocyte morphology[42] and are sufficiently immunosuppressive to hinder the proliferation of CD8⁺ T cells and IFN-γ expression[14]. However, it is not clear whether CD115⁻ M-MDSCs emerged from CD115⁺ cMoPs. If they are descendant cells of cMoPs, CD115 somehow needs to be downregulated in the BM. As CD115 is actively internalized by ligand binding,

**Fig. 4 CD115⁻ M-MDSCs preferentially differentiate into PMN-MDSCs outside the tumor mass. a–c** CFSE-labeled CD115⁻ or CD115⁺ M-MDSCs sorted from EL4 TB mice were injected intravenously into EL4 TB recipient mice (n = 3). Cells were preaped from recipient mice and pooled to analyze transplanted cells (CFSE⁺) by flow cytometry 2 days after injection. Data shown are from two biological replicates of each group. **a** Schematic diagram of experimental design. The letter "t" indicates that cells were isolated from TB mice. **b** Representative plot of transplanted CD115⁻ and CD115⁺ M-MDSC and ratios of PMN-MDSC (CD11b⁺Ly6G⁺Ly6Cˡᵒ) differentiation at indicated sites. **c** Representative plot and ratio of M2 TAM in the tumor. **d, e** CD115⁻ and CD115⁺ M-MDSCs were sorted and differentiated in vitro in supplemental GM-CSF (10 ng/ml) ± 5% EL4 TES for 6 days. Data pooled from two separated experiments (n = 3–6). Ratios of PMN-MDSC and M2 TAM (**d**) and representative H&E stained image (**e**) after in vitro culture. Arrows indicate PMN-MDSCs. Scale bars = 20 µm. Statistical comparisons were performed using multiple unpaired Student's t test: *p < 0.05. Data are mean or mean ± SD.

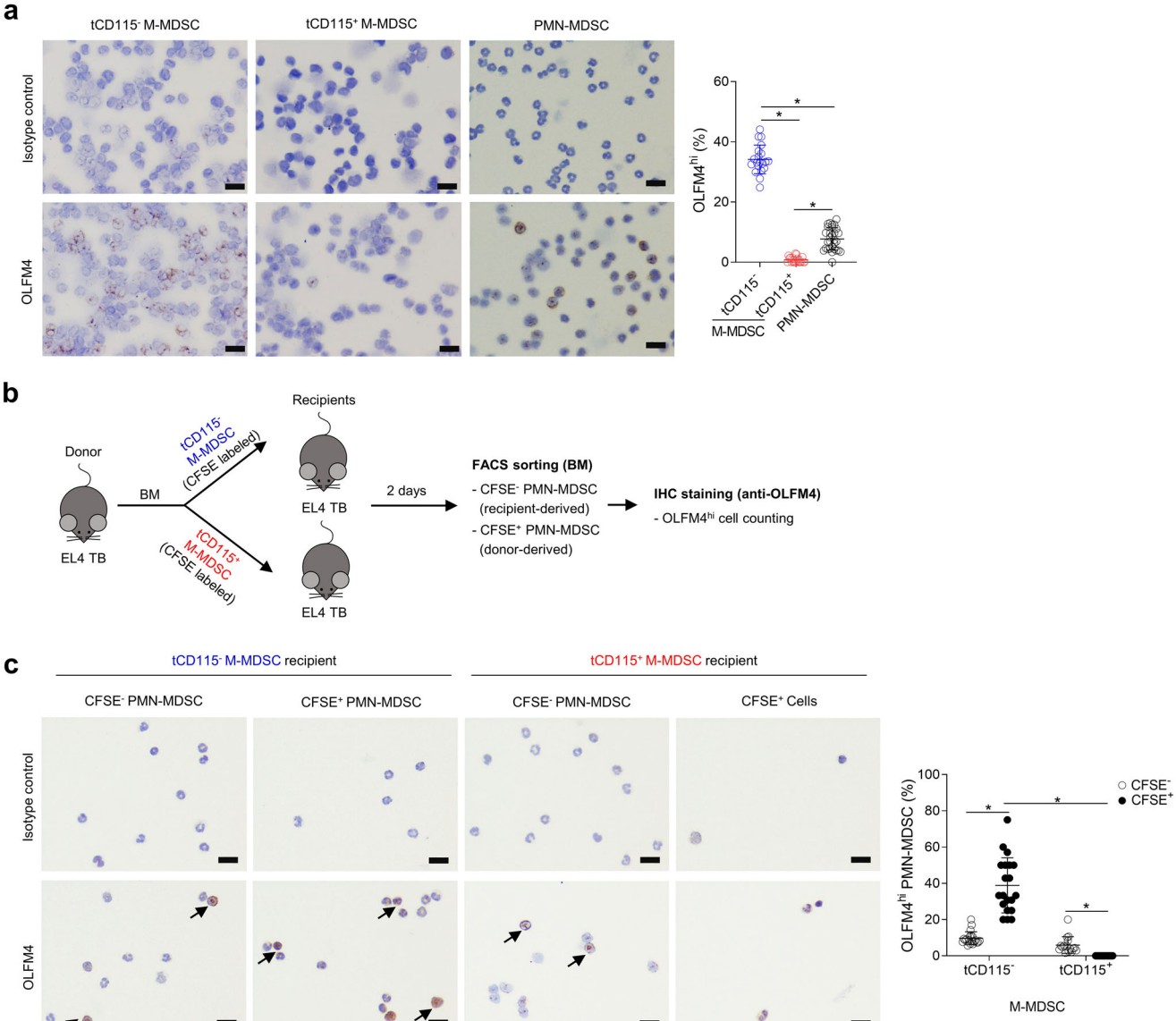

**Fig. 5 CD115⁻ M-MDSCs express OLFM4 and differentiate into OLFM4ʰⁱ PMN-MDSCs. a** BM cells from three EL4 TB mice were pooled and CD115⁻ M-MDSCs, CD115⁺ M-MDSCs, and PMN-MDSCs were sorted. Cells were stained for OLFM4 and OLFM4ʰⁱ cells were counted from ten random fields. Data pooled from two to three separate experiments. Representative IHC analysis of OLFM4 expression and ratios of OLFM4ʰⁱ cells. The letter "t" indicates that cells were isolated from TB mice. **b** Schematic diagram of experimental design. CD115⁻ and CD115⁺ M-MDSCs were sorted from donor EL4 TB mice and labeled with CFSE, then CD115⁻ and CD115⁺ M-MDSCs were injected into separate recipient EL4 TB mice (n = 3 per group). Two days later, BM cells from three recipient mice were pooled and OLFM4 expression in sorted PMN-MDSCs (CD11b⁺Ly6G⁺Ly6Cˡᵒ) were analyzed by IHC to determine the ratio of OLFM4ʰⁱ PMN-MDSC. **c** OLFM4ʰⁱ cells were counted from seven to ten random fields. IHC analysis of OLFM4 expression and ratios of OLFM4ʰⁱ cells. Data pooled from two separate experiments. Scale bars = 20 µm. One-way ANOVA with correction for multiple comparisons test was used: *p < 0.05. Data are mean ± SD.

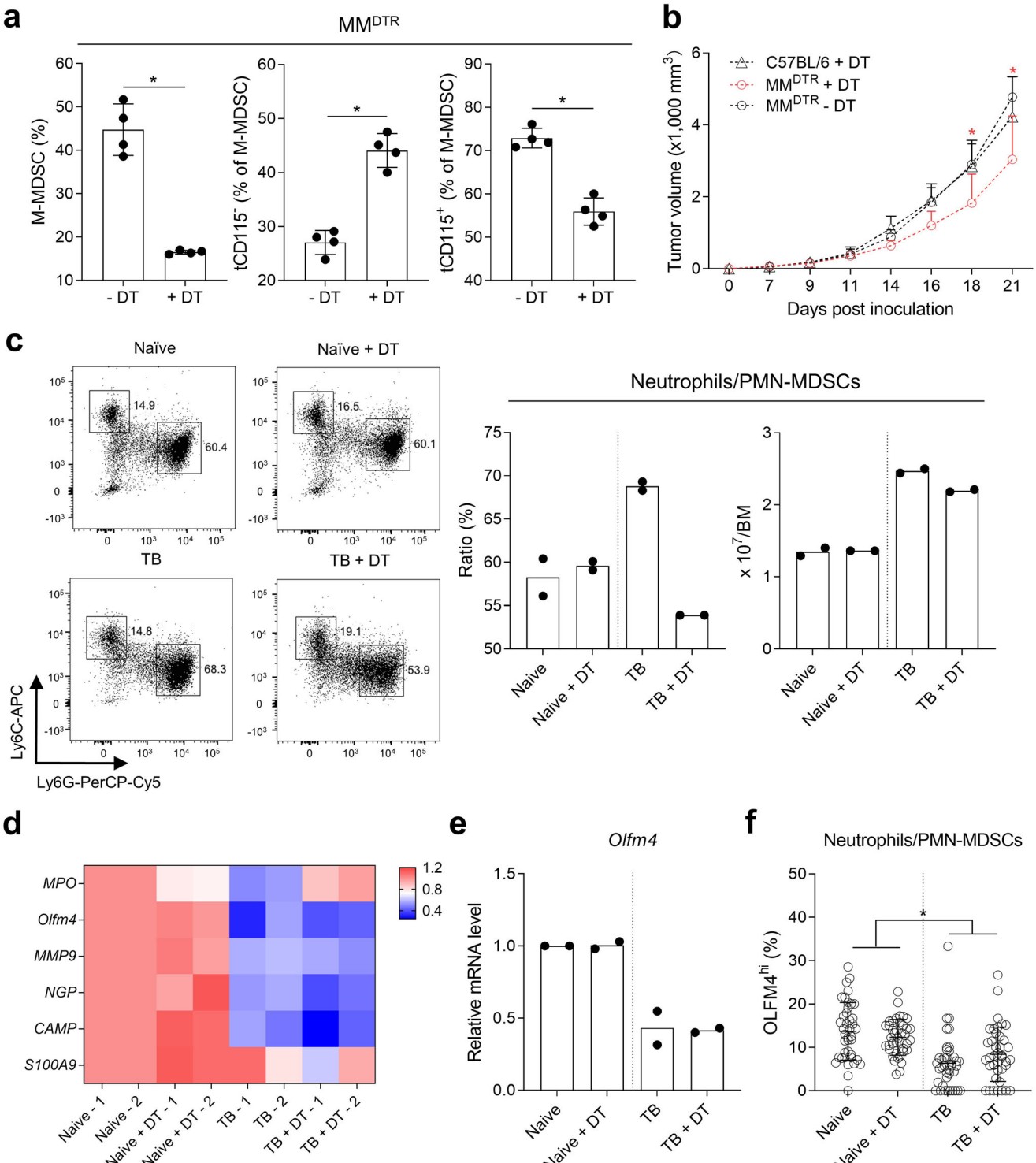

**Fig. 6 Depletion of CD115$^+$ M-MDSCs does not affect OLFM4$^{hi}$ PMN-MDSCs.** CD115$^+$ monocytic cells were transiently depleted in MM$^{DTR}$ mice by intraperitoneal injection of DT. **a** Ratio of M-MDSCs (% CD11c$^-$CD11b$^+$Ly6G$^-$SiglecF$^-$Ly6C$^{hi}$) and their subpopulations (CD115$^-$ and CD115$^+$) in BM of MM$^{DTR}$ mice with or without DT treatment ($n = 4$ per group). **b** Tumor volume of EL4 TB C57BL/6 and MM$^{DTR}$ mice with or without DT treatment ($n = 7$–9 per group). **c–f** BM cells were prepared from indicated mice ($n = 3$ per group) and neutrophilic cells (CD11b$^+$Ly6G$^+$Ly6C$^{lo}$) were sorted from pooled BM cells. **c** Representative plots of neutrophilic cells in BM of naïve and EL4 TB MM$^{DTR}$ mice with or without DT injection. Ratio and absolute numbers of neutrophilic cells in BM of naïve and EL4 TB mice with or without DT injection are shown in graphs. Data shown are from two biological replicates of each group. **d, e** Heatmap of neutrophil-associated gene expression (**d**) and *Olfm4* expression (**e**) in sorted neutrophilic cells from indicated mice. Data shown are from two biological replicates of each group. **f** IHC assessment of ratio of OLFM4$^{hi}$ neutrophilic cells in BM of indicated mice. OLFM4$^{hi}$ cells were counted from twenty random fields. Data pooled from two separate experiments. Statistical comparisons were performed using unpaired Student's $t$ test when only two groups were compared or by Bonferroni's test-corrected ANOVA when more than two groups were compared: *$p < 0.05$. Data are mean or mean ± SD.

bacterial DNA, and lipopolysaccharide, an extracellular factor (likely increased in the tumor host) may be required to maintain the CD115[−] subpopulation[43].

As myeloid progenitors share surface markers with the M-MDSCs used in this study for cell sorting, contamination of ancestry cells including granulocyte-monocyte progenitors (GMPs), which are Ly6C[+]CD115[−] and can generate neutrophils, should be avoided. To avoid contamination of granulocytes and myeloid progenitors, we gated out CD11b[+] cells that expressed CD11c, SiglecF, and Ly6G. Further, we gated Ly6C[hi] cells, as GMPs and granulocytes express less Ly6C (Ly6C[lo])[44,45]. We also examined the morphology of sorted M-MDSCs as well as their expression of CD135 (FLT3). While CD135 is an important surface marker of many types of dendritic cells, it is also expressed on the surface of GMPs, monocyte-dendritic cell progenitors (MDPs), and other progenitor cells[46,47]. Thus, we excluded progenitor contamination in CD115[−] & CD115[+] populations by checking CD135 expression. As expected, we did not find expression of CD135 on CD115[+] or CD115[−] M-MDSCs cells.

PMN-MDSCs, immunosuppressive myeloid cells found in the tumor host, outnumber M-MDSCs[14]. TB hosts have both suppressive PMN-MDSCs and non-suppressive PMN cells, making this population more diverse[5,6]. A recent study described new subset of PMN-MDSCs that highly express SiglecF. These cells accumulated in TB mouse lungs, but not in healthy lungs[48]. Also, another subset of PMN that expresses OLFM4 was reported in mice and humans[9,10]. In a study using OLFM4 knockout mice, OLFM4[+] and OLFM4[−] PMN showed similar phagocytic and transmigration activity. OLFM4[+] PMN expanded during experimental sepsis and an OLFM4 deficiency ameliorated the disease[9]. Similar to findings in mice, human OLFM4[+] and OLFM4[−] PMN displayed comparable bacterial phagocytosis and tissue transmigration as well as Fas ligand-induced spontaneous apoptosis[49]. Importantly, OLFM4[+] PMN was also increased during septic shock in humans[50]. These studies suggest that OLFM4[+] PMN is pathogenic in sepsis. The neutrophilic OLFM4[+] subset in cancer was investigated in a more recent study using mice in which OLFM4 was selectively deficient in myeloid cells[51]. When tested in a chemically induced colorectal cancer mouse model, loss of OLFM4 in PMN-MDSCs delayed the progression of disease. Mechanistically, OLFM4 interacts with LGALS3 to activate PMN-MDSCs through the NF-κB/PTGS2 pathway. Besides, other studies report that OLFM4 is a useful biomarker for predicting prognosis. OLFM4 is often overexpressed in normal tissue and early-stage tumors but downregulated in advanced tumor stages[52,53]. Low expression of OLFM4 promoted lymph node and distant metastasis resulting in a poor prognosis of triple-negative breast cancer patients and low expression of OLFM4 in early gastric cancer was associated with lymph node metastasis[54,55]. Previous studies have shown that OLFM4 contributes pro-inflammatory and pro-tumoral function of PMNs. However, OLFM4[hi] PMN-MDSCs may play contrary roles in other types of cancers due to different etiologies. Although neutrophils are derived from committed PMN progenitors, PMN-MDSCs can be generated from M-MDSCs but not from non-tumoral monocytes[38]. Thus, study of M-MDSC to PMN-MDSC differentiation is required when considering the potential importance of OLFM4[hi] PMN-MDSCs in tumorigenesis. Our study specifically demonstrates that the CD115[−] subset of M-MDSCs express higher levels of granulocytic-associated transcripts and can differentiate into OLFM4[hi] PMN-MDSCs at a higher rate.

CD115[−] M-MDSCs showed a series of gene expressions, including genes that code transcription factors and chemokine receptors. Of note, as a member of the CCAAT/enhancer-binding protein (C/EBP) family of transcription factors, Cebpe is important for secondary and tertiary granule formation in granulocytes[34]. Mutation of Cebpe in mice not only blocks the terminal differentiation and secondary granule formation, but also fails to produce functional neutrophils[56]. CXCR2 is essential for the release of neutrophils from the BM to circulation and CXCR2 deficiency retains neutrophils in the BM resulting in chronic neutropenia[57]. Transcriptional upregulation of neutrophil-related genes in M-MDSCs supports our finding that CD115[−] M-MDSCs are asymmetrically polarized by factors as yet unknown that make it distinguishable from CD115[+] M-MDSCs.

PMN-MDSCs have a very short life span in tumor mass and they are less populated in tumor mass than in the periphery[58], at least in part, due to tumor cells producing more chemokines for M-MDSC than PMN-MDSC recruitment[59]. Also, PMN-MDSCs are intolerant to low pH and hypoxic tumor microenvironments[60]. In line with the previous study findings by others, our data showed that CD115[−] M-MDSCs preferentially differentiated into TAM rather than remaining as M-MDSCs in TME. It is possible that the unique combination of cytokines that constitute TME strongly drive TAM differentiation. M1 TAM exists in perivascular regions with an anti-tumor function, and M2 TAM occupies hypoxia areas of the tumor with a pro-tumor function[61,62]. M-CSF is known to be involved in M2 TAM differentiation via CD115 signaling; however, we found that CD115[−] M-MDSCs can also differentiate to both M1 and M2 TAMs in vitro and in vivo. Therefore, we speculate there may be an alternative pathway that can differentiate both CD115[+] and CD115[−] M-MDSCs to M2 TAM independent of M-CSF, even though tumor mass usually provides a high level of M-CSF[63].

In conclusion, it will be important to clarify the functional role of CD115[−] M-MDSCs and their descendent cells. At this point we believe that the depletion of CD115[−] M-MDSCs from mice will be the best approach to define their role in vivo but in the current study we failed to find a specific target for CD115[−] M-MDSCs. As an alternative, we employed conditional knockout mice that can transiently deplete CD115[+] M-MDSCs and found that OLFM4[hi] PMN-MDSCs were not affected. The current findings indicate the need for studies to understand the complex heterogeneity of MDSC populations. In addition, CD115[−] monocytes in the steady-state condition need further investigation. Monocytes can differentiate into monocyte-derived macrophages (MMP) and DCs in non-tumoral diseases upon arrival at affected tissues[64]. This final differentiation to MMP is usually triggered by inflammatory responses and have a longer effect on disease progression than short-lived monocytes[64]. As we found that a significant portion of CD115[−] monocytes are present in non-tumor conditions, MMP that originate from CD115[−] M-MDSCs may play a unique role in inflammatory diseases.

## Methods

**Mice.** Wild-type C57BL/6 mice were purchased from Orient Bio (Gyeonggi-do, Korea). LysM-Cre (Lyz2[Cre]) and Csf1r[LsL-DTR] mice were purchased from Jackson Laboratory (Bar Harbor, ME, USA). The conditional knockout mice that express DTR-mCherry fusion protein in LysM-expressing cells (MM[DTR]) were generated by crossing Lyz2[Cre] and Csf1r[LsL-DTR] mice[65]. All mice were maintained under specific pathogen-free conditions at the Wide River Institute of Immunology (WRII) animal facility. Six- to 10-week-old male mice or female were used for experiments. The research was approved by the Institutional Animal Care and Use Committee at Seoul National University (190123-1).

**Mouse tumor model.** Mouse lymphoma (EL4), lung carcinoma (LLC1), and colon adenocarcinoma (MC38) cell lines were maintained using DMEM (HyClone Cytiva, Marlborough, MA, USA) supplemented with 10% fetal bovine serum (FBS) (Invitrogen, Carlsbad, CA, USA) and 1% penicillin/streptomycin (Invitrogen) at 37 °C in an incubator containing 5% $CO_2$. Mice were subcutaneously injected with one of three cell lines, EL4 ($5 \times 10^5$ cells), LLC1 ($5 \times 10^5$ cells), or MC38 ($1.5 \times 10^6$ cells), and sacrificed when the tumor reached 2.0 cm.

**Single-cell preparation.** According to the experiment design, single-cell suspensions were prepared from desired organs. In brief, the BM single-cell suspension was flushed out of femurs. The harvested spleen was teased apart into a single-cell suspension with a syringe plunger. We prepared single-cell suspensions from the lung and liver with Lung Dissociation Kit (Miltenyi Biotec, Nordrhein-Westfalen, Germany) and Liver Dissociation Kit (Miltenyi Biotec), respectively, per the manufacturer's instructions. To analyze tumor-infiltrating immune cells, tumor mass was dissociated with a tumor dissociation kit (Miltenyi Biotec) following the manufacturer's instructions. After obtaining single cells from different organs, erythrocytes were eliminated with red blood cell (RBC) lysis buffer. Single blood cells were generated from directly lysed blood with RBC lysis buffer. For CD115 expression analysis, cold EDTA buffer was used for BM and blood cell preparation[66]. Cells were ready for use after passage through a 70-μm nylon cell strainer.

**Flow cytometry.** Cells were incubated with CD16/CD32 Fc blocking antibody (2.4G2, 1:100; BD Biosciences, San Jose, CA, USA) for 10 min, then stained with fluorochrome-conjugated antibodies for 30 min on ice. CD11c (N418, 1:200), Ly6G (1A8, 1:300), CD115 (AFS98, 1:300), CD206 (C068C2, 1:400), F4/80 (BM8, 1:300), MHC II (M5/114, 1:300), Siglec-F (E50-2440, 1:300), and 7-AAD Viability Staining Solution were purchased from BioLegend (San Diego, CA, USA). CD11b (M1/70, 1:400), Ly6C (AL-21, 1:200), CD8a (53-6.7, 1:200), CD3 (17A2, 1;300), DAPI solution, and Annexin V were purchased from BD Biosciences. We performed flow cytometry with an LSRFortessa flow cytometer and sorted cells with an FACSAria III cell sorter (both from BD Biosciences). For data analysis, we used FlowJo software version 10.4 (Ashland, OR, USA).

**RNA preparation and RNA-Seq analysis.** Total RNA was extracted from cells using the RNeasy Mini Kit (QIAGEN, Hilden, Germany) and residual DNA was removed with a DNA-free Kit (Thermo Fisher Scientific, Waltham, MA, USA) according to the manufacturer's instructions. Samples were loaded onto a 1% agarose gel before electrophoresis. DNA-free conditions were confirmed by SYBR Gold staining (Invitrogen). The quantity and the quality of RNA were evaluated by Nanodrop 2000 spectrophotometer (Thermo Fisher Scientific) and bioanalyzer (Agilent 2100; Agilent Technologies, Santa Clara, CA, USA). RNA-Seq library preparation and sequencing were performed by Macrogen (Korea). Libraries were prepared with 1 μg of total RNA from two biological replicates of each group using TruSeq Stranded mRNA LT Sample Prep Kit and analyzed by Illumina Sequencer (both from Illumina, San Diego, CA, USA). The differentially expressed genes were selected based on a two-fold change and false discovery rate (FDR) < 0.05.

**T cell suppression assay.** $CD8^+$ T cells were purified from spleen of naive C57BL/6 mice using Dynabeads FlowComp Mouse CD8 Kit (Thermo Fisher Scientific). $CD8^+$ T cells were labeled with 0.5 μM of CFSE (Thermo Fisher Scientific) and seeded at a density of $4 \times 10^4$ in a 96-well plate in complete RPMI 1640 medium (GE Healthcare Life Sciences, MA, USA) containing 10% FBS and 1% penicillin/streptomycin. Mouse T-Activator CD3/CD28 microbeads (Thermo Fisher Scientific) were added to culture medium at a 1:1 ratio. Sorted $CD115^-$ M-MDSCs or $CD115^+$ M-MDSCs from EL4 TB mice were co-cultured with $CD8^+$ T cells at ratios of 2:1, 1:1, 0.5:1, 0.25:1, and 0:1. Cells sorted from naive mice were used as a negative control for the assay. The CFSE dilutions in $CD8^+$ T cells were analyzed by flow cytometry 3 days after co-culture. Two independent experiments were performed. Cell culture supernatant was collected and stored at −80 °C before interferon (IFN)-γ was evaluated by enzyme-linked immunosorbent assay.

**In vitro phagocytic activity assay.** Single-cell suspensions from BM or blood of naive or EL4 TB mice were plated at densities of $5 \times 10^5$ cells/well (BM) or $1 \times 10^6$ cells/well (blood) in 48-well plates. FITC-conjugated *E. coli* (Cayman, Ann Arbor, MI, USA) was incubated with cells for 1 h at 37 °C with 5% $CO_2$. Loosely bound *E. coli* was removed using a trypan blue quenching solution and stained with relevant fluorochrome-conjugated antibodies as shown in Supplementary Fig. 1a, then analyzed by flow cytometry.

**Hydrogen peroxide ($H_2O_2$)-induced apoptosis assay.** Single cells from BM of naive or EL4 TB mice were suspended at a density of $2 \times 10^6$ cells/well in complete RPMI medium containing different concentrations of $H_2O_2$ (0, 50, 100, 200, 400 μM). After 2 h of incubation at 37 °C, cells were thoroughly washed with cold PBS and stained with relevant fluorochrome-conjugated antibodies (Supplementary Fig. 1a). Apoptotic cells were stained with FITC-conjugated Annexin V and the proportions of early ($AnnexinV^+7ADD^-$) and late ($AnnexinV^+7ADD^+$) apoptotic cells were analyzed by flow cytometry.

**In vitro M-MDSC differentiation.** Purified BM $CD115^-$ and $CD115^+$ M-MDSC from EL4 or LLC1 TB mice were cultured in a 12-well plate at a density of $1.5 \times 10^5$ cells/well. RPMI complete medium supplemented with 1 mM sodium pyruvate (Gibco, Thermo Fisher Scientific, Waltham, MA, USA), 55 μM 2-Mercaptoethanol (Gibco), and 10 ng/ml GM-CSF (PeproTech, Rocky Hill, NJ, USA) was used as a base medium. To mimic the tumor microenvironment, EL4 or LLC1 tumor explant

supernatants (TES) were prepared according to the previous study[67]. Briefly, one gram of EL4 or LLC1 tumor was minced and cultured in RPMI complete medium for 24 h and supernatant was collected and stored at −80 °C until use. For M-MDSC differentiation, 5% EL4 or LLC1 TES was added to the base medium. On day 3, the supernatant from TES free wells was collected as a condition media (CM) for migration and invasion assays. After collecting supernatant, each well was refilled with new media and cultured for an additional three days. Cells were collected for flow cytometric analysis after 6 days of culture. Some cells were separately cultured in Cell Imaging Coverglass (Eppendorf, Hamburg, Germany) and stained with hematoxylin-eosin (H&E, Merck Millipore, Burlington, MA, USA) for microscopic imaging.

**Enzyme-linked immunosorbent assay (ELISA).** Levels of cytokines (TGF-β, VEGF, and IL-10) in CM and IFN-γ in the co-cultured supernatant were evaluated with commercially available ELISA kits (R&D Systems, Minneapolis, MN, USA) according to the manufacturer's instructions.

**Migration and invasion assay.** The migration of tumor cells (EL4 or LLC1) was evaluated using Transwell permeable supports (Corning, NY, USA). The upper chamber of an 8-μm transwell insert was seeded with equal numbers of tumor cells ($1 \times 10^5$) in serum-free medium. Then one of the following media were added into the lower chamber, medium supplemented with or without 10% FBS, medium with 10% FBS and 10 ng/ml GM-CSF, CM medium prepared from in vitro $CD115^-$ or $CD115^+$ M-MDSC culture fluid. Cell migration was examined after incubation for 30 h at 37 °C with 5% $CO_2$. EL4 cells that migrated into the lower chambers were collected and stained with 0.4% trypan blue solution for counting by microscopy. LLC1 cells that migrated to the other side of the membrane were fixed with 70% ethanol and stained with 0.2% crystal violet. After a wash with sufficient distilled water, excess crystal violet was removed, and the cell-bound crystal violet was eluted with 33% acetic acid. The absorbance at 590 nm of the eluent was measured using a plate reader. A known number of LLC1 cells was cultured, stained, and quantified in the same manner as described above in order to convert absorbance to cell number. Invasion assays were performed using the same protocol, except that Matrigel (Corning) was applied to the transwell insert. The migrated or invaded cells were calculated as a percentage of migration or invasion (migrated cell number or invaded cell number / initial cell number × 100%).

**Quantitative RT-PCR.** Single strands of cDNA were synthesized by reverse-transcription of total RNA (1 μg) using Maxime RT Premix Kit (iNtRON Bio-technology, Seongnam-si, Korea). RT-PCR was performed using the SYBR-Green PCR Master Mix on a QuantStudio 6 Flex Real-time PCR system (both from Applied Biosystems, Waltham, MA, USA). All analyses were done in triplicate, and the expression of the target genes was normalized to *Gapdh*. Relative expression ratios were calculated by the comparative $2^{-\Delta\Delta Ct}$ method[68]. All primers used in this study are listed in Supplementary Table 1.

**Immunohistochemistry.** Sorted cells ($2 \times 10^4$) were spun on slides using Cytospin 4 (Thermo Fisher Scientific) and stained per the basic protocol described in the manual of the Mouse and Rabbit Specific HRP/DAB (ABC) Detection IHC Kit (Abcam, Cambridge, United Kingdom). In brief, slides were incubated overnight with a rabbit anti-OLFM4 antibody (Novus, Littleton, CO, USA) or normal rabbit IgG (Santa Cruz, CA, USA) at a 1:300 dilution at 4 °C in a humidity chamber. After being developed with the DAB chromogen, all slides were counterstained with hematoxylin (Merck Millipore). We counted $OLFM4^{hi}$ cells that exhibit dense and ring-like OLFM4 signals by randomly selecting 10 or more fields in each glass slide[9,10].

**In vivo tracking of M-MDSCs.** Purified $CD115^-$ or $CD115^+$ M-MDSCs from BM of TB mice were labeled with CFSE (Thermo Fisher Scientific) and suspended in DPBS. Recipient TB mice were injected intravenously with either $4 \times 10^6$ $CD115^-$ or $CD115^+$ CFSE-labeled M-MDSCs two weeks after tumor implantation. The injected cells ($CFSE^+$) were analyzed by flow cytometry 48 h after transplantation.

**$CD115^+$ M-MDSC depletion.** $MM^{DTR}$ mice were implanted with $2.5 \times 10^5$ EL4 cells. Each mouse received one intraperitoneal injection (10 ng/g of body weight) of diphtheria toxin (DT) (List Biological Laboratories, Campbell, CA, USA) on days 7, 9, and 11 after tumor implantation (Supplementary Fig. 7b). Time intervals (every 48 h) of DT injection were optimized by flow cytometry blood analysis (Supplementary Fig. 7a). After sacrifice on day 14 after tumor implantation, single cells were prepared for flow cytometry analysis. $CD11b^+Ly6G^+Ly6C^{lo}$ cells from naive or TB mice were sorted for OLFM4 immunohistochemistry (IHC) and qRT-PCR analysis. C57BL/6 mice were used as a negative control to exclude effects related to DT.

**Statistical and reproducibility.** Statistical analysis was performed using GraphPad Prism software (version 9.0; GraphPad, La Jolla, CA, USA). A two-tailed Student's *t* test with a 95% confidence interval was used for comparison between two groups. One-way ANOVA or two-way ANOVA test with Bonferroni correction for

multiple comparison was used in statistical analysis with more than two groups. All data are presented as mean (SEM) or mean ± SD, and significance was defined as $p < 0.05$. The sample size, number of independently performed experiments, and statistical methods are indicated in the figure legends.

**Reporting summary**. Further information on research design is available in the Nature Portfolio Reporting Summary linked to this article.

## Data availability

The RNA-Seq analysis of CD115− monocytes and CD115+ monocytes from naive and TB mice data have been deposited to the Gene Expression Omnibus (GEO) datasets (https://www.ncbi.nlm.nih.gov/geo/) with the dataset identifier GSE199008. The source data generated during the study are available in Supplementary Data 1.

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

## Acknowledgements

The authors thank Dr. Je-In Youn for useful discussion and tumor cell lines. This work was supported by grants from National Research Foundation of Korea (NRF) grant funded by the Korea government (MSIT) 2021R1F1A1059826, 2022R1F1A1074547 (SUS), and the financial support of the Catholic Medical Center Research Foundation in the program year of 2020 (SUS).

## Author contributions

N.K, S.-Y.S., and S.-U.S. conceived and supervised the study. Y.Z. and S.-U.S. wrote the manuscript and Y.Z. performed experiments.

## Competing interests

The authors declare no competing interests.
