## [Peer Review File · Communications Biology]

Reviewers' comments:

Reviewer #1 (Remarks to the Author):

In this manuscript the authors studies MDSC, specifically M-MDSCs to characterize the difference between CD115+ and CD115- M-MDSCs. They used flow cytometry, functional studies, RNAseq and adoptive transfer to characterize these cell types. Some of their primary findings include the identification of a CD115- M-MDSC population. The authors characterize the population relative to CD115+ population. They show that the CD115- population gives rise to a greater proportion of OLFM4+ neutrophils, where the CD115+ population does not. The manuscript includes a large amount of data. Figures are well organized. Legends are well written. While there is important data presented here, I have some concerns about how the conclusions were drawn.

Major Concern-

While I am very familiar with MDSCs, I am not an expert on their ontogeny. The authors study CD115- seemingly assuming they are M-MDSCs and no other cell types. Particularly, in the bone marrow, is it not possible at that the CD115- subpopulation also includes other early stage myeloid progenitors? Specifically, GMP are Ly6C positive and CD115 negative. If GMP cells are included in the authors CD115- populations, it would be expected that they give rise to neutrophils and monocytes. Could a different interpretation of the data be that tumor injection stimulates myelopoiesis, which would result in an increase in CD11b, Ly6c+ CD115- GMP in the bone marrow and some spill over into the blood and thus these cells are no M-MDSCs at all. I believe the authors need to better define the Cd11b+Ly6C+ CD115- population.

Minor Concerns:

-The authors note they only used male mice for these studies. Current guidelines from the NIH and many journals is to include both sexes equally.

-Data on OLFM4 expression in neutrophils suggests that neutrophils are closer to OLFM4 low and OLFM4 high rather than OLFM4 negative and OLFM4 positive. Can the authors please clarify how they defined positive and negative to better understand their percentage OLFM4 positive calculations. For example, the last panel of figure 4a, there are some neutrophils that are clearly positive and others that are quite weak.

Reviewer #2 (Remarks to the Author):

Here Zou et al. examine the transition of a CD115 null subset of MDSC to OLIFM4+ PMN-MDSC. The data are, for the most part, well presented and convincing. The data suggest that the CD155- subset has a distinct physiology and developmental track from the CD155+ subsets. The limitation to the work is the fact that the role of the OLIFM4+ PMN-MDSC in cancer is currently unknown. It is shown that the CD115- M-MDSC are slightly better inhibitors than the CD115+ M-MDSC. Based on previous literature it is assumed that the OLIFM4+ PMN-MDSC that are derived from the CD115- population are qualitatively different from OLIFM4- PMN-MDSC. This literature suggests that OLIFM4 is associated with sepsis and expressed in certain tumor types. However, how the OLIFM4+ PMN-MDSC population differs functionally is unclear.

Specific comments on the data:

The differences in T cell suppression between CD155+ and - M-MDSC (fig 2b) is very small. Is this relevant in disease?

The peroxide studies suggest that 50% of the CD115+ population is apoptosis + before the treatment (fig 2e). What is going on here? This is not commented on in the text.

The comparisons used in the sequencing data are not clear to this reviewer. The 1290 and 1107 DEG of fig 3c are the DEG between naive and tumor 115- and 115+ cells, respectively, correct? Where are the 1723 DEG assessed in 3d derived from? What is the comparison that led to this?

Reviewer #3 (Remarks to the Author):

In this study, Zou et al reported that CD115 expression could be used to identify 2 subsets of M-MDSCs. CD115 (-) M-MDSCs increased in bone marrow and blood as tumors progressed. They further showed that CD115 (-) and CD115 (+) M-MDSCs were functionally distinct, and CD115 (-) M-MDSCs displayed higher ability to differentiate into PMN-MDSCs than CD115 (+) M-MDSCs. Both in vivo and in vitro CFSE tracking assay showed that CD115 (-) M-MDSCs were able to differentiate into both OLFM4 (+) and OLFM4 (-) PMN-MDSCs, whereas CD115 (+) M-MDSCs differentiated into OLFM4 (-) PMN-MDSCs. Overall, the study is well-designed and the findings are interesting. The data were clearly presented and explained. There are some concerns regarding the significance of these observations:

1. Whether there is difference between CD115 (-) and CD115 (+) M-MDSCs in cytokine profiling? such as immunosuppressive IL-10, TGF-beta.
2. What is the impact of CD115 (-) and CD115 (+) M-MDSCs on tumor growth and tumor metastasis? The authors may need to transfer CD115 (-) and CD115 (+) M-MDSCs during the induction of tumor bearing models, to see whether there is difference in tumor growth and tumor metastasis.
3. What is the functional differences between OLFM4 (+) and OLFM4 (-) PMN-MDSCs? What is the significance of CD115 (-) and CD115 (+) M-MDSCs in the differentiation into OLFM4 (+) and OLFM4 (-) PMN-MDSCs. The authors need to explain and discuss their observations.
4. What signaling or tumor-derived soluble factors drive the generation of CD115 (-) and CD115 (+) M-MDSCs?

Reviewer #1

In this manuscript the authors studies MDSC, specifically M-MDSCs to characterize the difference between CD115+ and CD115- M-MDSCs. They used flow cytometry, functional studies, RNAseq and adoptive transfer to characterize these cell types. Some of their primary findings include the identification of a CD115- M-MDSC population. The authors characterize the population relative to CD115+ population. They show that the CD115- population gives rise to a greater proportion of OLFM4+ neutrophils, where the CD115+ population does not. The manuscript includes a large amount of data. Figures are well organized. Legends are well written. While there is important data presented here, I have some concerns about how the conclusions were drawn.

Major Concern-

While I am very familiar with MDSCs, I am not an expert on their ontogeny. The authors study CD115- seemingly assuming they are M-MDSCs and no other cell types. Particularly, in the bone marrow, is it not possible at that the CD115- subpopulation also includes other early stage myeloid progenitors? Specifically, GMP are Ly6C positive and CD115 negative. If GMP cells are included in the authors CD115- populations, it would be expected that they give rise to neutrophils and monocytes. Could a different interpretation of the data be that tumor injection stimulates myelopoiesis, which would result in an increase in CD11b, Ly6c+ CD115- GMP in the bone marrow and some spill over into the blood and thus these cells are no M-MDSCs at all. I believe the authors need to better define the Cd11b+Ly6C+ CD115- population.

Reply: We thank the reviewer for this important point. It is true that ancestry myeloid cells express the same surface markers as monocytes. In our original submission, we tried to avoid possible contamination of granulocytes (especially neutrophils and eosinophils) and progenitors during the monocyte analysis. First, we gated out SiglecF- and Ly6G-expressing CD11b⁺ cells (Supplementary figure 1a). Second, we gated Ly6C^{hi} cells because Ly6C expression is lower (Ly6C^{int}) in granulocytes and their progenitor cells, including GMP (PMIDs: 27381735, 27043410). To further make sure our monocyte gating was free of progenitors, we checked the morphology of sorted CD11b⁺Ly6C^{hi}CD115⁻ cells. Both sorted CD115⁺ and CD115⁻ CD11b⁺Ly6C^{hi} cells had the typical morphology of monocytic cells. If we had contamination of progenitor cells (such as GMPs), many of the sorted cells should have shown morphologic characteristics of Ly6C^{int} cells (i.e., generally smaller in cell size, round and regular in shape, and circular or donut-shaped nuclei, Appendix figure 1).

Appendix figure 1. BM CD11b⁺Ly6C^{int} and CD11b⁺Ly6C^{hi} cells were sorted and placed on separate slides. Cells were H&E stained and observed by microscopy.

To further confirm that monocytic cells were used in our experiments, we checked CD135 (FLT3) expression on CD11b⁺Ly6C^{hi}CD115⁻ and CD11b⁺Ly6C^{hi}CD115⁺ cells. While CD135 is an important surface marker of many types of dendritic cells, it is also expressed on the surface of GMP, monocyte-dendritic cell progenitors (MDPs) and other progenitor cells (PMID: 29875383, 24613914). When CD135 expression on both CD115⁺ and CD115⁻ CD11b⁺Ly6C^{hi} cells from naïve and TB mice were tested, no CD135 expression was detected (Appendix figure 2).

Appendix figure 2. CD135 (FLT3) expression of BM CD11c⁻Ly6G⁻SiglecF⁻CD11b⁺Ly6C^{hi} cells. Plots and histograms show representative data from three individual experiments with similar results.

In addition, a previous study also reported that sorted splenic M-MDSCs can differentiate into PMN-MDSCs when cultured *in vitro* or transplanted into mice, indicating not only granulocytic progenitors but also that certain CD11b⁺Ly6C^{hi} cells can differentiate into PMN-MDSCs (PMID: 23354483). Overall, we believe that the sorted CD11b⁺Ly6C^{hi}CD115⁻ cells used in our study did not have significant contamination of early stage myeloid progenitors and was sufficiently potent to differentiate into PMN-MDSCs. In addition, CD11b⁺ cells in PBMCs contain fewer Ly6C^{int} cells. Therefore, monocytic populations (Ly6C^{hi}) are easy to sort without contamination and the chance of Ly6C^{int} progenitor cells is very low (lines 447-457).

Minor Concerns:

-The authors note they only used male mice for these studies. Current guidelines from the NIH and many journals is to include both sexes equally.

Reply: We appreciate the reviewer's comment. Male mice are known to develop stronger disease phenotypes in certain mouse models such as DSS-induced colitis and diet-induced NAFLD (PMIDs: 25962374, 30924946). We also used male mice in this study because most cancers affect males more than females (PMID: 32295632). Although we agree with the reviewer's concern that both sexes need to be tested, we also had to comply with our institution's ethical guidelines that recommend using the

fewest animals possible. Although we couldn't repeat every experiment using female mice, we did compare the ratio of CD115⁻ cells from M-MDSC populations of both sexes. As a result, we found comparable ratios of CD115⁻ cells in both male and female tumor-bearing mice (Appendix figure 3). In addition, a previous study using female mice also showed that M-MDSCs can be differentiated into PMN-MDSCs (PMID: 23354483). Therefore, we believe that our CD11b⁺Ly6C^{hi}CD115⁻ M-MDSCs would similarly differentiate in female mice. We discuss this potential concern about possible results due to sex differences in the text of the revised manuscript (lines 275-276).

Appendix figure 3. BM cells were isolated from male and female TB mice and the ratio of CD115⁻ cells from whole M-MDSCs (CD11c⁻Ly6G⁻SiglecF⁻CD11b⁺Ly6C^{hi}) were calculated.

-Data on OLFM4 expression in neutrophils suggests that neutrophils are closer to OLFM4 low and OLFM4 high rather than OLFM4 negative and OLFM4 positive. Can the authors please clarify how they defined positive and negative to better understand their percentage OLFM4 positive calculations. For example, the last panel of figure 4a, there are some neutrophils that are clearly positive and others that are quite weak.

Reply: We agree with the reviewer's observation that weak OLFM4 signals are detected in "OLFM4 negative" neutrophils. Similar weak OLFM4 signals were shown in a previous study using isolated human neutrophils. In that study, the authors divided neutrophils into OLFM4^{low} and OLFM4^{high} populations (PMID: 22187488). Although it is not clear whether this weak OLFM4 signal represents marginal OLFM4 expression or is due to experimental conditions, it seems reasonable to change "negative" to "lo" and "positive" to "hi" (i.e., OLFM4^{lo} and OLFM4^{hi} populations) in our study. Based on previous reports (PMIDs: 22187488, 30537894), we counted OLFM4^{hi} when cells exhibited dense and ring-like OLFM4 signals. We have changed the terminology throughout our revised manuscript and now explain how we define high and low expression of OLFM4 (lines 229-230).

Reviewer #2

Here Zou et al. examine the transition of a CD115 null subset of MDSC to OLFM4+ PMN-MDSC. The data are, for the most part, well presented and convincing. The data suggest that the CD115⁻ subset

has a distinct physiology and developmental track from the CD155+ subsets. The limitation to the work is the fact that the role of the OLIFM4+ PMN-MDSC in cancer is currently unknown. It is shown that the CD115- M-MDSC are slightly better inhibitors than the CD115+ M-MDSC. Based on previous literature it is assumed that the OLIFM4+ PMN-MDSC that are derived from the CD115- population are qualitatively different from OLIFM4- PMN-MDSC. This literature suggests that OLIFM4 is associated with sepsis and expressed in certain tumor types. However, how the OLIFM4+ PMN-MDSC population differs functionally is unclear.

Specific comments on the data:

The differences in T cell suppression between CD155+ and – M-MDSC (fig 2b) is very small. Is this relevant in disease?

Reply: We appreciate the reviewer's comment. We agree that the difference in T cell suppression between CD115+ and CD115- M-MDSCs is quite small. We expected CD115- M-MDSCs to suppress T cell proliferation from the initial stage of this study because this subset was already part of M-MDSCs in many previous cancer studies based on surface markers (CD11b⁺Ly6C^{hi}). Figure 2 clearly confirms that our novel subset can be functionally classified as "suppressor cells"; however, it was still important to note the degree of T cell suppression was significantly different from the CD115+ counterpart.

So far, our best characterization of CD115- M-MDSCs is that this cell has higher granulocytic gene expression compared to CD115+ M-MDSCs and is a potent precursor of OLFM4^{hi} PMN-MDSCs (we now divide PMN-MDSC into OLFM4^{lo} and OLFM4^{hi} following Reviewer 1's suggestion). We agree with the reviewer's concern that the disease relevance of CD115- M-MDSCs is unclear because the role of OLFM4^{hi} PMN-MDSCs has yet to be well defined. We plan to continue functional analysis of CD115- M-MDSCs to see if they can be used as a predictor of cancer prognosis.

The peroxide studies suggest that 50% of the CD115+ population is apoptosis + before the treatment (fig 2e). What is going on here? This is not commented on in the text.

Reply: We exposed CD115+ and CD115- M-MDSCs to different concentrations of hydrogen peroxide (H₂O₂) and measured the ratio of apoptotic cells 2 hours after treatment. We found ~50% of CD115+ monocytes underwent apoptosis in the group without H₂O₂ treatment and considered these cell deaths as "spontaneous apoptosis" during culture. Previous studies also showed that *in vitro*-generated (mouse bone marrow-derived) MDSCs and human monocytes can induce spontaneous apoptosis (PMID: 21368871, 34712241). Importantly, spontaneous apoptosis was induced at different degrees depending on cell type, ranging from 5% to 40%. Our data show that CD115+ M-MDSCs from TB mice are more prone to induce spontaneous apoptosis (~40% to 50%) compared to CD115- M-MDSCs (~20%). Similar results were obtained from naïve mice, where CD115+ monocytes were more susceptible to H₂O₂-induced apoptosis than CD115- monocytes. We have included this explanation in the revised manuscript (lines 297-301).

The comparisons used in the sequencing data are not clear to this reviewer. The 1290 and 1107 DEG of fig 3c are the DEG between naïve and tumor 115⁻ and 115⁺ cells, respectively, correct? Where are the 1723 DEG assessed in 3d derived from? What is the comparison that led to this?

Reply: We compared tumor CD115⁺ and CD115⁻ M-MDSCs in see Figure 3c. Analysis of four groups showed 1723 DEGs (i.e., nCD115⁺, nCD115⁻, tCD115⁺, tCD115⁻, where n and t represent naïve and tumor, respectively). We drew four possible combinations of Venn diagrams for two sets. We put the tCD115⁺ vs. tCD115⁻ in the main figure as we thought this comparison was of the most interest. Other Venn diagrams are shown in Supplementary figure 4a.

Reviewer #3

In this study, Zou et al reported that CD115 expression could be used to identify 2 subsets of M-MDSCs. CD115⁻ M-MDSCs increased in bone marrow and blood as tumors progressed. They further showed that CD115⁻ and CD115⁺ M-MDSCs were functionally distinct, and CD115⁻ M-MDSCs displayed higher ability to differentiate into PMN-MDSCs than CD115⁺ M-MDSCs. Both in vivo and in vitro CFSE tracking assay showed that CD115⁻ M-MDSCs were able to differentiate into both OLFM4⁺ and OLFM4⁻ PMN-MDSCs, whereas CD115⁺ M-MDSCs differentiated into OLFM4⁻ PMN-MDSCs. Overall, the study is well-designed and the findings are interesting. The data were clearly presented and explained. There are some concerns regarding the significance of these observations:

1. Whether there is difference between CD115⁻ and CD115⁺ M-MDSCs in cytokine profiling? such as immunosuppressive IL-10, TGF-beta.

Reply: We appreciate the reviewer's comment. We isolated CD115⁺ and CD115⁻ M-MDSCs from EL4 and LLC TB mice and measured TGF- β secretion in the culture medium. CD115⁺ M-MDSCs from EL4 TB mice showed more TGF- β secretion than CD115⁻ M-MDSCs but the difference was not statistically significant (Appendix figure 4). Moreover, TGF- β secretion was comparable for both CD115⁺ and CD115⁻ M-MDSCs in LLC1 TB mice. We attempted to measure IL-10 and VEGF but were under the detection limit. We included this data in Supplementary Figure 3g and described that suppressive cytokine production (i.e., TGF- β) is comparable between the two M-MDSC subsets (lines 312-321).

Appendix figure 4. TGF- β secretion by CD115⁺ and CD115⁻ M-MDSCs. CD115⁺ and CD115⁻ M-MDSCs were isolated from EL4 or LLC1 TB mice and cells were cultured for 3 days. TGF- β concentrations were measured from conditioned medium (CM).

2. What is the impact of CD115 (-) and CD115 (+) M-MDSCs on tumor growth and tumor metastasis? The authors may need to transfer CD115 (-) and CD115 (+) M-MDSCs during the induction of tumor bearing models, to see whether there is difference in tumor growth and tumor metastasis.

Reply: We thank the reviewer for suggesting the cell transfer experiment. We had considered doing adoptive transfer of CD115⁺ and CD115⁻ M-MDSCs into TB mice but didn't perform this study for two reasons. First, the pathophysiological contribution of transferred CD115⁺ and CD115⁻ M-MDSCs may be weak because recipient mice should already have both CD115⁺ and CD115⁻ M-MDSCs. Therefore, we thought the depletion of either CD115⁺ and CD115⁻ M-MDSCs might be a more reasonable way to see their function. We could deplete CD115⁺ M-MDSCs using MM^{DTR} mice and concluded that CD115⁺ M-MDSCs do enhance tumor growth, however, we could not find an efficient way to selectively deplete CD115⁻ M-MDSCs. Second, even if we want to transfer CD115⁺ or CD115⁻ M-MDSCs, large numbers of cells are needed for injection. According to our calculations, we can isolate about 5×10^5 cells from one TB mouse without accounting for loss during the FACS sorting process. Thus, isolation of enough CD115⁻ M-MDSCs would require large numbers of mice and take a long time for the sorting process, which in turn would worsen the cell conditions.

Instead, we tried to determine the role of CD115⁺ and CD115⁻ M-MDSCs in tumor metastasis using *in vitro* transwell experiments. We prepared conditioned medium (CM) from CD115⁺ and CD115⁻ M-MDSC culture and added them to the lower chambers. Both CM from CD115⁺ and CD115⁻ M-MDSCs significantly increased the EL4 and LLC1 cell migration and invasion but no difference was observed between CD115⁺ and CD115⁻ CM. Although we couldn't demonstrate the functional difference between CD115⁺ and CD115⁻ M-MDSCs, it was important to see that CD115⁻ MDSCs can enhance tumor migration and invasion (Appendix figure 5). We include this result as Supplementary figure 3f (lines 304-311).

Appendix figure 5. Tumor migration (a) and invasion (b) assay performed in the presence of CD115⁺

and CD115⁻ M-MDSC conditioned medium (CM). Both EL4 and LLC1 cancer cell lines were tested.

3. *What is the functional differences between OLFM4 (+) and OLFM4 (-) PMN-MDSCs? What is the significance of CD115 (-) and CD115 (+) M-MDSCs in the differentiation into OLFM4 (+) and OLFM4 (-) PMN-MDSCs. The authors need to explain and discuss their observations.*

Reply: Functional differences between OLFM4⁺ and OLFM4⁻ neutrophils/PMN-MDSCs were suggested in some previous research, and we are also interested in doing future follow-up studies for functional characterization of OLFM4^{hi} PMN-MDSCs. (Per the suggestion of Reviewer 1, in our revised manuscript we now divide PMN-MDSCs into OLFM4^{lo} and OLFM4^{hi}.) Our current study focuses on the characterization of a novel subset of M-MDSCs and the ancestry role of OLFM4^{hi} PMN-MDSCs rather than on functional analysis. The physiological meaning of CD115⁻ MDSC differentiation into OLFM4^{hi} PMN-MDSCs is unclear as the role of OLFM4⁺ expressing PMN-MDSCs is not well defined. Nevertheless, we agree that it is important to explain the significance of CD115⁻ MDSC differentiation into OLFM4^{hi} PMN-MDSCs.

Recently, Chen et al. showed that OLFM4 expression in myeloid cells contributes to PMN-MDSC activation in a mouse colorectal cancer model (PMID: 35487976). As the cancer model used in the study by Chen et al. is inflammation-driven tumorigenesis and OLFM4⁺ PMN-MDSCs are reported to be associated with inflammatory diseases, OLFM4^{hi} PMN-MDSCs may have different roles in other types of cancer due to different etiologies. Therefore, while our study suggests a pro-tumoral role of OLFM4 in myeloid cells, we were unable to determine the function of OLFM4^{hi} PMN-MDSCs. In the Discussion section of our revised manuscript we now discuss the impact of OLFM4^{hi} PMN-MDSCs in our model (lines 469-473 and 478-481).

4. *What signaling or tumor-derived soluble factors drive the generation of CD115 (-) and CD115 (+) M-MDSCs?*

Reply: We thank the reviewer for the comment. Finding a signal or soluble factors that drive generation of two different M-MDSC populations would be challenging but is obviously important. As we can also find CD115⁻ subpopulations in naïve mice, generation of two subsets might not be tumor-specific. As we discuss in our manuscript, we do not have evidence that CD115⁻ M-MDSCs are differentiated from common monocyte progenitors.

REVIEWERS' COMMENTS:

Reviewer #1 (Remarks to the Author):

The authors have made significant effort in addressing the concerns I raised on initial review. While I am still very curious about heterogeneity within the CD11b, Ly6c+ CD1115- population, I believe the authors have adequately address my overall concern.

Reviewer #2 (Remarks to the Author):

This reviewer has no further comments.

Reviewer #3 (Remarks to the Author):

The authors have properly addressed my points.